# *HD-Zip III* Gene Family: Identification and Expression Profiles during Leaf Vein Development in Soybean

**DOI:** 10.3390/plants11131728

**Published:** 2022-06-29

**Authors:** Jing Gao, Jiyu Chen, Lingyang Feng, Qi Wang, Shenglan Li, Xianming Tan, Feng Yang, Wenyu Yang

**Affiliations:** 1College of Agronomy, Sichuan Agricultural University, Huimin Road 211, Wenjiang District, Chengdu 611130, China; 2020201020@stu.sicau.edu.cn (J.G.); 2020201038@stu.sicau.edu.cn (J.C.); lingyang.feng@pku-iaas.edu.cn (L.F.); 2020201016@stu.sicau.edu.cn (Q.W.); 2019201025@stu.sicau.edu.cn (S.L.); 2019301094@stu.sicau.edu.cn (X.T.); mssiyangwy@sicau.edu.cn (W.Y.); 2Sichuan Engineering Research Center for Crop Strip Intercropping System, Chengdu 611130, China; 3Key Laboratory of Crop Ecophysiology and Farming System in Southwest, Ministry of Agriculture, Chengdu 611130, China

**Keywords:** soybean, *HD-Zip III* gene family, leaf vein, auxin, brassinosteroid

## Abstract

Leaf veins constitute the transport network for water and photosynthetic assimilates in vascular plants. The class III homeodomain-leucine zipper (*HD-Zip III*) gene family is central to the regulation of vascular development. In this research, we performed an overall analysis of the *HD-Zip III* genes in soybean (*Glycine max* L. Merr.). Our analysis included the phylogeny, conservation domains and *cis*-elements in the promoters of these genes. We used the quantitative reverse transcription-polymerase chain reaction to characterize the expression patterns of *HD-Zip III* genes in leaf vein development and analyze the effects of exogenous hormone treatments. In this study, twelve *HD-Zip III* genes were identified from the soybean genome and named. All soybean HD-Zip III proteins contained four highly conserved domains. *GmHB15-L-1* transcripts showed steadily increasing accumulation during all stages of leaf vein development and were highly expressed in cambium cells. *GmREV-L-1* and *GmHB14-L-2* had nearly identical expression patterns in soybean leaf vein tissues. *GmREV-L-1* and *GmHB14-L-2* transcripts remained at stable high levels at all xylem developmental stages. *GmREV-L-1* and *GmHB14-L-2* were expressed at high levels in the vascular cambium and xylem cells. Overall, *GmHB15-L-1* may be an essential regulator that is responsible for the formation or maintenance of soybean vein cambial cells. *GmREV-L-1* and *GmHB14-L-2* were correlated with xylem differentiation in soybean leaf veins. This study will pave the way for identifying the molecular mechanism of leaf vein development.

## 1. Introduction

Leaf veins provide mechanical support and are at the forefront of nutrient and water transport in photosynthesis and transpiration [1,2]. Leaf veins consist of the xylem, phloem, and cambium, which are highly organized within the vascular bundle. Cells in the cambium can differentiate to form either the phloem or the xylem. Then, the xylem and phloem are further expanded via the differentiation of cells derived from divisions in the cambium [3]. Leaves develop an adaxial side specialized for light capture, and an abaxial side specialized for gas exchange. Leaf veins are typically positioned where the adaxial and abaxial domains meet. The xylem is present in the adaxial position, and the phloem is present in the abaxial position [3,4]. The characteristics of leaf veins are directly or indirectly related to the capacity for photosynthetic carbon fixation, water absorption, and anti-interference of plants, and are thus critical factors in adapting to adverse stress [5,6]. The optimization of leaf vasculature is essential for the efficient performance of crop plants [7]. Therefore, studying the molecular mechanism of vein development is of great theoretical significance and application value. The *HD-Zip III* gene family regulates almost all the vascular developmental fate decisions in this process, including vascular specification, patterning, and differentiation [8,9].

*HD-Zip III* family members have overlapping and antagonistic roles in controlling vascular development. Five *HD-Zip III* genes have been found in Arabidopsis (*Arabidopsis thaliana*): *REV*, *PHB/ATHB9*, *PHV/ATHB14*, *ATHB8*, and *ATHB15/CAN* [10]. *PHB* and *PHV* perform overlapping functions with *REV* in controlling the abaxial–adaxial patterning of organs [11]. The *REV* mutant causes a reduction in interfascicular xylem fibers [12,13]. The double mutants *REV PHB* and *REV PHV* enhance the vascular defects of the *REV* mutant, specifically, vascular bundles with remarkably few lignified cells [10]. Loss-of-function *KANADI* mutants exhibit vascular patterning defects opposite to that of the rev mutant, increasing the number of xylem cells [14]. *HD-Zip III* and *KANADI* have opposing roles in ad/abaxial of the organ formation. Once vascular are formed, *HD-Zip III* and *KANADI* are required to coordinate adaxial and abaxial growth [15]. The histological analysis of the *ATHB8* single mutant revealed a regular vascular system without morphological alterations [16]. A functional redundancy might explain the lack of the evident phenotypes of *ATHB8* mutants within the *HD-Zip III* family. Slight perturbations in vascular development are seen in *CNA* single mutant stems; i.e., the vascular bundles are frequently poorly distributed around the stem periphery [10]. Several pieces of evidence indicate mutual antagonism among the *REV*, *CNA*, and *ATHB8* genes during vascular development. The *CNA* and *ATHB8* mutations suppress the *REV* phenotype. The lignification of xylem tissue and interfascicular fibers are restored in *ATHB8 CNA REV* triple mutants [10].

Auxin is the primary signal involved in the ontogeny of the vascular system [17]. *HD-Zip III* is a regulatory factor in the regulation of vascular development by auxin (Brandt et al., 2012). Cambium formation is promoted by high auxin levels activating *HD-Zip III* transcription factors in Arabidopsis [18]. Previous studies have proposed the auxin-flow canalization hypothesis, which states that *HD-Zip III* genes form an integrated feedback loop along with auxin, the auxin polar transporter *PIN*, and the auxin response factor *(MP/ARF)* [19,20]. The onset of *ATHB8* expression is directly and positively regulated by *MP* through an auxin response element in the *ATHB8* promoter and is followed by the induction of the generation and maintenance of procambial cells.

Previous studies on various mutants of vascular patterns have revealed the involvement of another new player, brassinosteroids (BRs). BR is a key regulator of the xylem, by acting to stimulate cambium cell differentiation [21]. Over expression of the BR biosynthesis gene showed increased xylem formation [22]. Brassinazole (BRZ), a specific inhibitor of BR biosynthesis, inhibits the differentiation of tracheary cells from cambium cells; such suppression is reversed by the addition of BRs [23,24]. *HD-Zip III* genes function in zinnia (*Zinnia elegans* L.) vascular differentiation in response to BR signaling. The expression of *ZeHB-12* and *ZeHB-10* (*REV* homologous transcripts) coincides with the differentiation of cambium cells into tracheary cells. The expression of *ZeHB10* and *ZeHB12* is repressed by BRZ but is rapidly induced by BR [25].

Although considerable evidence indicates that the *HD-Zip III* gene family is the essential family in controlling stem and root development, the details of individual *HD-Zip III* genes in the development of leaf veins have not been resolved. The structure of the leaf veins directs the mobilization of photosynthates from source to sink, and the optimization of leaf vein architecture is essential for the efficient performance of crop plants [7]. Soybean (*Glycine max* L. Merr.) is one of the most important crops worldwide and produces substantial amounts of proteins and oils [26]. Genome-wide investigations on the *HD-Zip* gene family in soybean have been reported, and gene expression in soybean roots under dehydration and salt stress has been studied [5,27]. Updating the *HD-Zip III* gene family in soybean has become possible with the release of a new version of the soybean genome. In this study, we aimed to identify the *HD-Zip III* gene family in the soybean genome and analyze their phylogenetic relationship, structural characteristics, and *cis*-element regions. Then, we investigated the functions of *HD-Zip III* genes during leaf vein development in soybean.

## 2. Results

### 2.1. Identification of GmHD-Zip III Genes and Analysis of Basic Physicochemical Properties

In this work, twelve *HD-Zip III* genes were found in the soybean genome and named based on their orthologous in Arabidopsis (Appendix A). According to sequence orthologous, 12 *GmHD-Zip III* genes were divided into three groups. The largest group was Group 1, which contained six *GmHD-Zip III* genes; Group 2 and 3 had two and four genes, respectively (Figure 1). We also identified interspecific orthologous genes. The orthologous gene pairs were GmHB8-L-1\GmHB8-L-2, GmHB15-L-1\GmHB15-L-2, GmHB15-L-3\GmHB15-L-4, GmHB14-L-1\GmHB14-L-2, GmHB14-L-3\GmHB14-L-4, and GmREV-L-1\GmREV-L-2 (Appendix A). As shown in Appendix A, a total of 60 *HD-Zip III* homologous proteins were identified from five legumes. The phylogenetic tree showed that HD-Zip III proteins had a high degree of homology across all of the investigated plants (Figure 1).

Chromosomes 4, 5, 6, 9, 11, 12, and 15 contained only one *GmHD-Zip III* gene, whereas chromosome 7 contained two *GmHD-Zip III* genes. Chromosome 8 exhibited the highest density of *GmHD-Zip III* genes (Appendix A).

The results demonstrate that the *HD-Zip III* proteins have lengths of 838–853 amino acids and an average length of 844 amino acids. The theoretical isoelectric point value of GmREV-L-1 was the lowest, and that of GmHB14-L-3 was the highest. Protein hydrophobic and hydrophilic analysis showed that soybean *HD-Zip III* proteins were hydrophilic and unstable. The instability coefficients were between 45.74 and 50.47. Molecular weight analysis revealed that GmHB14-L-4 (93 958.15 Da) had the maximum protein molecular weight and that GmREV-L-1 (92 062.93 Da) had the minimum molecular weight (Appendix A). Under the predicted subcellular localization of *GmHD-Zip III*, all of the genes in this family were located in the nucleus (Appendix A).

### 2.2. Gene Structure, Conservative Domain, and Motif Analyses

We analyzed the conserved motifs of amino acid sequences (Appendix A). The results showed that all members of the *HD-Zip III* family contained almost the same motifs (1–10). Several had functional implications. For example, motif 6 specified the HD domain, motif 1 corresponded to the Zip domain, and motifs 12 and 14 represented the methionine–glutamic–lysine–histidine–leucine–alanine (MEKHLA) domain. Some motifs, such as 2, 4, and 3, specified the START domain. As revealed by the analysis of genetic structure, *HD-Zip III* gene structures were remarkably conserved. Each *GmHD-Zip III* gene had precisely 18 exons (Figure 2b). Exon length and exon–intron patterns were generally conserved.

All of the 12 GmHD-Zip III proteins contained four highly conserved domains (Figure 2a), namely, START, MEKHLA, HD, and Zip. The closely related HD and Zip domains are the structural basis of this family of proteins, which together determine their functions as transcription factors. The START domain was predicted to be a lipid or steroid-binding receptor that can bind to small hydrophobic molecules, such as phospholipids and steroids, and is highly conserved in evolution [28]. The *HD-Zip III* family genes also contain an additional highly conserved MEKHLA domain. The MEKHLA domain is unique to the *HD-Zip III* family [29]. However, studies on the potential role of the MEKHLA domain in plants are limited. The MEKHLA domain shares high similarity with the Per-Arnt-Sim (PAS) domain [30]. The PAS domain might have originated from algae, and may be involved in the regulation of light signals received by plants and related to photosynthesis [31].

### 2.3. cis-Elements in GmHD-Zip III Promoters

*cis*-Acting elements play an important role in modulating the molecular switches of dynamic transcriptional regulation in response to developmental processes and hormonal signaling [32]. On the basis of their functions, the *cis*-acting elements were grouped into several classes: stress, development, and hormone-responsive elements (Appendix A, Appendix A). All soybean *HD-Zip III* genes contained ARF elements, which are putative binding sites for auxin response factor (ARF/MP) proteins [33]. ARF proteins may bind ARF elements to activate or repress transcription of *GmHD-Zip III* genes (Figure 3).

### 2.4. Expression Patterns of GmHD-Zip III Genes under Hormone Treatment

BRs are positive regulators of xylem differentiation, and BRZ, a specific inhibitor of BR biosynthesis, suppresses xylem differentiation [34]. We analyzed the effects of BRs and BRZ on the accumulation of *GmHD-Zip*
*III* transcripts to investigate the relationships between the expression of these genes and BRs. Five *GmHD-Zip*
*III* genes increased expression after BR treatment (Appendix A). *GmHB8-L-1*, *GmHB14-L-1, GmHB14-L-2*, *GmREV-L-1*, and *GmHB15-L-2* all showed significantly upregulated expression levels. *GmHB15-L-2* mRNA accumulation exhibited the greatest change. Expression analysis showed that BRZ strongly suppressed the accumulation of the transcripts of *GmHB14-L-1*, *GmHB14-L-2*, and *GmREV-L-1* (Figure 4b). The other seven genes showed no significant changes. These results strongly suggest that the expression of *GmHB14-L-1, GmHB14-L-2*, and *GmREV-L-1* can be induced or promoted by endogenous BRs. Our study also verified the promoting effect of BRs on xylem differentiation in leaf veins. The areas of xylem precursor cells, xylem cells, and lignin content in soybean leaf veins were significantly increased after 10 days of BR treatment (Figure 5a,b).

Auxin is involved in the regulation of vascular tissue development [35]. Treatment with a polar auxin transport inhibitor can mimic the vascular defect caused by *HD-Zip III* mutations [36]. Treatments with exogenous IAA and the auxin polar transport inhibitor NPA were applied to test whether these genes play an essential role in auxin-regulated vascular development. IAA significantly induced *GmHB15-L-1*, *GmHB15-L-2*, *GmHB15-L-3*, *GmHB15-L-4*, and *GmHB14-L-4* (Appendix A), but did not induce changes in the other seven genes. *GmHB15-L-2*, *GmHB14-L-4*, and *GmHB15-L-4* were not severely suppressed by NPA. NPA significantly inhibited the expression of *GmHB15-L-1* and *GmHB15-L-3* (Figure 4b). No absolute correlation was observed between the number of ARF elements and expression level. Although all *GmHD-Zip III* genes had ARF-elements, seven genes were not induced by auxin at the transcriptional level (Appendix A). These results provided support for the idea that auxin is the initiator of *GmHB15-L-1* and *GmHB15-L-3.* The areas of leaf vein cambium cells, xylem percursor cells, and xylem cells were significantly increased after 10 days of IAA treatment (Figure 5a,b).

### 2.5. Expression of GmHD-Zip III Genes in Leaf Vein Development in Soybean

We analyzed the changes in the gene expression of *GmHD-Zip III* in samples obtained at different leaf vein developmental stages to understand the possible functions of *GmHD-Zip III* genes in leaf vein development in soybean. Figure 6a shows the transverse sections of leaf veins taken at five different soybean developmental stages. The results revealed that *GmHB15-L-1* and *GmREV-L-1* were very specifically expressed in vascular tissues at all stages, including the 0, 12, 24, 48, and 96 h stages, of vascular maturation (Figure 6a). *GmHB15-L-1* was the earliest expressed gene in leaf vein development (Figure 5b). The first group genes (*GmHB15-L-1*, *4*) exhibited similar expression patterns. As the leaf vein developed, the relative expression levels of the two genes gradually decreased. *GmREV-L-1* and *GmHB14-L-2* transcripts steadily accumulated during 12 to 48 h of development. As shown in Figure 6a, these time points corresponded to key xylem developmental stages. Subsequently, we examined gene expression in the cambium and the developing xylem and phloem. *GmHB15-L-1* was expressed at low levels in the developing xylem and phloem but was highly expressed in the vascular cambium (Figure 7). *GmREV-L-1* and *GmHB14-L-2* were expressed at low levels in the developing phloem but expressed at high levels in the vascular cambium and xylem (Figure 7). The distinctive expression patterns of these genes in leaf veins imply their functional diversity in association with vascular development. These results suggest that *GmHB15-L-1*, *GmREV-L-1*, and *GmHB14-L-2* play essential roles in leaf vein development.

## 3. Discussion

### 3.1. Phylogeny of GmHD-Zip III Genes Reflects Their Functional Conservation in Soybean

*HD-Zip* is a vital transcription factor family and exists only in the plant kingdom. HD-Zip transcription factors can be generally classified into the I, II, III, and IV subfamilies in accordance with their conserved sequences [37]. The majority of the reports available on *HD-Zip I* subfamilies respond to abiotic stresses and are crucial for maintaining plant growth under unfavorable environments [11]. The *HD-Zip II* gene family regulates the shade-avoiding mechanism during the photosynthetic process [38]. Genes in *HD-Zip IV* have well-characterized functions in trichome formation and epidermal cell differentiation [39,40]. Members of the third subfamily (*HD-Zip III*) control leaf polarity, embryogenesis, and vascular development [41]. The purpose of this study was to investigate the role of the *HD-Zip* gene family in soybean vein development. so we focused on the HD-Zip *III* subfamily rather than other subfamilies. In this work, 12 *HD-Zip III* genes were identified in the current version of the soybean genome (Figure 1). A previous work identified 11 *Gm**HD-Zip*
*III* genes in the soybean genome (v1.01, JGI Glyma1.0) and investigated their responses to salt and dehydration stress [27]. Compared with the study of Belamkar et al., our study identified one more *HD-Zip III* member, namely, GLYMA_12G075800. All of the soybean *HD-Zip III* proteins that we identified belonged to the *HD-Zip III* family because they contained the four conserved domains of the *HD-Zip III* transcription family, START, MEKHLA, HD, and Zip. The difference between our results and the findings of Belamkar et al. may have been due to the use of the different released versions of the soybean genome. Eight *HD-Zip III* genes have been identified in poplar [42], four in barley [43], six in medicago [44], five in rice [45], four in strawberry [46], and five in cucumber [47]. The *HD-Zip III* gene family in soybean is by far more prevalent than in other plant species. The large number of *GmHD-Zip III* genes in soybean could have resulted from whole-genome duplication events. The soybean genome is believed to have undergone at least two independent duplications from a diploid ancestor to yield the actual polyploid plant [48].

The *HD-Zip III* gene family is very conserved. All of the identified *HD-Zip III* genes include four conserved domains [43,47,49,50]. In tomatoes and cotton, the locations and lengths of the motifs in the *HD-Zip III* protein sequences were also completely conserved [49,50]. The gene structures of the members of the soybean *HD-Zip III* family were highly similar in terms of exon and intron numbers and conserved protein motifs (Figure 2a,b). This high similarity further supported the reliability of the phylogenetic relationship analysis. The results of the phylogenetic analyses showed a high degree of the conservation of *HD-Zip III* protein sequences from different species. *HD-Zip III* genes may have similar functions in different species due to their conserved structure. Recent analyses on Arabidopsis, tomato, rice, and poplar have indicated that the *HD-Zip III* family is a key transcriptional regulator of vascular development and is highly expressed in the vascular tissues of roots, stems, and leaves [10,42,45,50]. The tissue-specific expression profiles obtained in this work suggested that soybean *HD-Zip III* genes were specifically expressed in the root, stem, and leaf (Appendix A). The first group was specifically expressed in leaves and stems. The expression level of the third group in roots was significantly higher than that in other sites. The *GmHD-Zip III* gene family is specifically expressed in different tissues with various subfamilies, and it may be involved in vascular development in certain sites.

### 3.2. Auxin Activating GmHB15-L-1 in the Regulation of Soybean Leaf Veins’ Cambium Development

The *HD-Zip III* family is at the center of a complex network required for initiating and maintaining plant vascular tissues [51]. However, such information is limited to roots and stems, and the general roles of *HD-Zip III* genes in the development of soybean leaf veins are still unknown. Despite the great variety of patterns in vascular systems, a common mechanism likely underlies the regulation of vascular tissue formation. In our study, *GmHB15-L-1* was found to be a key player in the regulation of cambium formation in soybean leaf veins.

Considerable evidence indicates that auxin is the essential signal in promoting vascular cambium zone development. Decapitation of seedlings prevents the auxin supply from reaching the shoots and represses cambium activity, while application of exogenous IAA restores cambium activity [52]. Our results confirmed that auxin promoted the formation of the leaf vein cambium and showed that the area of leaf vein cambium cells increased significantly after 10 days of exogenous auxin treatment (Figure 5). In Arabidopsis, *ATHB8* is a regulatory factor involved in the regulation of vascular cambium cell development by auxin [53]. The auxin element in the ATHB8 promoter is required for both ATHB8 procambial expression and auxin inducibility [20]. *GmHB8-L-1* and *GmHB8-L-2* had ARF elements, but these genes were not induced by auxin (Figure 3 and Appendix A). In poplar, *PtrHB8* is highly expressed in the vascular cambium and developing xylem tissue [42]. This expression pattern is different from that of *GmHB8-L-1,2* in soybean. The expression level of *GmHB8-L-1,2* was low during vein development (Figure 5a). A likely candidate for regulating the cambium cell development is *GmHB15-L-1*, which is most similar to *GmHB8-L-1*. First, the expression of *GmHB15-L-1* was significantly induced by IAA. The expression level of *GmHB15-L-1* was significantly inhibited under treatment with the auxin polar transport inhibitor NPA (Figure 4a). Second, the expression pattern results revealed that *GmHB15-L-1* expression levels were related to leaf vein development and showed steadily increasing transcript accumulation during all stages of leaf vein development (Figure 6b). The heat map data revealed that *GmHB15-L-1* was the earliest expressed gene during vascular development (Figure 7). Third, *GmHB15-L-1* was highly expressed in the vascular cambium. *ZeHB-13*, a homolog of the *ATHB15* gene in zinnia, is localized preferentially in cambium cells. Histochemical promoter analysis using *ATHB15::GUS* transgenic Arabidopsis indicated that, consistent with the expression patterns of *GmHB15-L-1* in soybean, *ATHB15* is active specifically in the cambium [54]. Overall, these results strongly suggest that a member of the *HD-Zip III* family, namely, *GmHB15-L-1*, may be an essential transcriptional regulator responsible for cambial tissue formation or maintenance.

The positive feedback loop auxin-flow–*MP*–*HD-Zip III*–*PIN1*–auxin-flow is typical in cambial cell development, a crucial mechanism in determining vascular cell fates [55]. The auxin response factor *MP* directly bound the auxin response element in the promoter sequence of *ATHB8*, thereby regulating *ATHB8* expression [20]. *ATHB8* is required to stabilize *PIN1* expression against auxin transport perturbation in cambial cells, resulting in the formation and maintenance of procambial cells [16,19]. We found that GmHB15-L-1 had ARF elements and was significantly induced by IAA, which is highly expressed in the vascular cambium. Soybean *PIN1* genes show the same expression pattern in the cambial cell [56]. Whether the conserved pathway also appears in soybean remains to be clarified.

### 3.3. Soybean HD-Zip III Genes Perform Overlapping Functions in Promoting Xylem Differentiation in Leaf Veins

Previous studies have linked BRs to vascular development. BR treatment promotes poplar stem growth and xylem formation [21]. In cultured zinnia cells, the levels of BR increase drastically prior to tracheary cell differentiation, and this increase is indispensable for progressing to the last stage of xylem cell differentiation (Yamamoto et al., 1997). The role of BR during soybean leaf veins development is not as well explored. After BR treatment, the areas of xylem cells and lignin content in leaf veins were significantly increased (Figure 6a,b). Our results are consistent with previous reports in poplar and zinnia that point to BRs promoting the differentiation of xylem cells. *HD-Zip III* functions in xylem cell differentiation by responding to BR signaling [51,54]. The xylem cell-specific accumulation of *ZeHB10* (the homologous gene of *ATHB8*) mRNA has also been observed in xylogenic cell culture [57]. The expression of *ZeHB11* and *ZeHB12* (*REV* homologous transcripts) is repressed by BRZ but is rapidly induced by BR [25]. Although *HD-Zip III* have been suggested to be involved in xylem cell development, the details of individual *HD-Zip III* functions in soybean leaf vascular differentiation have not been clarified. We characterized the expression patterns of *HD-Zip III* genes in leaf vein differentiation and analyzed the effects of BR and BRZ on the accumulation of *GmHD-Zip III* transcripts.

On the basis of the findings shown here, *GmREV-L-1* and *GmHB14-L-2* play pivotal roles in the differentiation of the xylem. Expression analysis showed that *GmHB14-L-2* and *GmREV-L-1* expression was induced by BR but was suppressed strongly by BRZ (Figure 4b). In zinnia, the rapid induction of *ZeHB12* occurs upon BR treatment. Consistent with that of *GmREV-L-1* in soybean, the expression of *ZeHB12* is repressed by BRZ. We used qRT-PCR to check the expression patterns of *GmREV-L-1* and *GmHB14-L-2* in leaf veins at different developmental stages (Figure 6a). Our results showed that *GmREV-L-1* and *GmHB14-L-2* were expressed in all of the examined stages. Throughout these stages, the relative transcript levels of *GmREV-L-1* and *GmHB14-L-2* were stably maintained at high levels in leaf veins at 12, 24, and 48 h of development (Figure 6b). As shown in Figure 5a, these time-points corresponded to vital developmental stages of the xylem. In Arabidopsis, *ATHB14/PHV* and *REV* exhibit overlapping expression in the adaxial domains in vascular bundles [58]. The *REV PHV* double mutant enhances the vascular defects of the *rev* mutant, i.e., vascular bundles with remarkably few lignified cells [10]. Consistent with a previous work, this work showed that *GmREV-L-1* and *GmHB14-L-2* had nearly identical expression patterns in soybean vascular tissues and were primarily expressed in vascular cambium cells and developing xylem cells (Figure 7). *GmHB14-L-2* may perform overlapping functions with *GmREV-L-1* in vascular cell differentiation. These facts suggest that *GmREV-L-1* and *GmHB14-L-2* play pivotal roles in xylem differentiation.

Given the relationship between BRs and *HD-Zip III* genes, the START domain deserves future investigation. The START domain has lipid binding capability [59]. In plants, the START domain is predominantly found in HD-Zip gene family. Lipid/sterol ligands can directly modulate HD-Zip transcription factor activity [60,61]. Consequently, soybean *HD-Zip III* genes may be activated through the binding of sterols or lipids to their START domain. Given that BRs are the initiators of soybean *HD-Zip III* genes, they may be candidates for binding to the START domain and act as pivotal signals that activate proteins.

## 4. Materials and Methods

### 4.1. Identification and Phylogenetic Tree Construction

To identify the *GmHD-Zip*
*III* genes in soybean, we used the sequences of five *Arabidopsis HD-Zip III* proteins [10] downloaded from the TAIR website (http://www.arabidopsis.org/ (accessed on 10 November 2021) as query sequences in the TBLASTP searches against the soybean genome (Glycine_max_v4.0), on the NCBI database (https://www.ncbi.nlm.nih.gov/genome/ (accessed on 10 November 2021) [62]. All sequences with an e-value below 10^−10^ were candidate soybean *HD-Zip III* proteins. Twelve *GmHD-Zip III* proteins were initially isolated in this research. Next, we performed BLASTP separately using each of the 12 *HD-Zip III* proteins as a query sequence in the soybean genome database. This process was repeated until no new *HD-Zip III* protein was found. The PLAZA database was used to obtain the orthology gene pairs for soybean (https://bioinformatics.psb.ugent.be/plaza/versions/plaza_v4_5_dicots/ (accessed on 16 June 2022)) [63]. We identified the homologous proteins of *HD-Zip III* in another four legumes, namely, chickpea (*Cicer arietinum*), peanut (*Arachis hypogaea*), kidney bean (*Phaseolus vulgaris*), and wild soybean (*Glycine soja*). Five Arabidopsis *HD-Zip III* proteins were input as query sequences into BLASTP searches with the barrel medic genome (MtrunA17r5.0-ANR), chickpea genome (A SM33114v1), peanut genome (arahy. Tifrunner. gnm1.KYV3), kidney bean genome (PhaVulg1_0), and wild soybean genome (A SM419377v2) in the NCBI database. We downloaded the protein sequences of *HD-Zip III* family genes that were newly identified in recent years in six plants, including strawberry (*Fragaria vesca*) [46], cucumber (*Cucumis sativus* L.) [47], poplar [42], tomato [50], medicago (*Medicago truncatula*) [44], and rice (*Oryza sativa*) [64].

The amino acid sequences of *HD-Zip III* were aligned using MEGA7.0 software. Phylogenetic analysis was constructed using the neighbor-joining method and 1000 bootstraps [65].

### 4.2. Chromosomal Mapping and Analysis of Basic Physical and Chemical Properties

All *GmHD-Zip III* genes were mapped on the 20 chromosomes of soybean using gene annotation (Glycine_max_v4.0_genomic.gff) acquired from NCBI by TBtools software [35]. The molecular characteristics of the soybean *HD-Zip III* proteins, including the theoretical predictions of isoelectric point and molecular weight, were analyzed using ExPASy Protparam Tool (https://web.expasy.org/compute_pi/ (accessed on 7 November 2021) [36]. Plant-mPLoc (http://www.csbio.sjtu.edu.cn/bioinf/plant-multi/ (accessed on 1 November 2021) was used to predict the subcellular localizations of the *HD-Zip III* proteins.

### 4.3. Gene Structure, Conserved Domains, and Motif Identification

Intron/exon structures were downloaded from the NCBI genome database. We used the NCBI Conserved Domains Database (CDD) (https://www.ncbi.nlm.nih.gov/Structure/bwrpsb/bwrpsb.cgi/ (accessed on 10 November 2021) to predict the conserved domains of the soybean *HD-Zip III* proteins [66,67]. MEME-MAST programs (http://meme.nbcr.net/meme/meme.html (accessed on 15 November 2021) were used to predict the conserved protein motifs, with the motif length set to 6–200 and the *e* value to <1 × 10^−10^ [68].

### 4.4. Plant Transcription Factor Binding Sites in the Promoters

For this study, the 2000 bp region upstream of the annotated transcription start site for each gene was evaluated for promoter motifs [66]. TBtools was utilized to extract *GmHD-Zip III* genes’ CDS (Coding Sequences) in the promoter sequences and visualization mapping. We used Plantcare (http://bioinformatics.psb.ugent.be/webtools/plantcare/html/ (accessed on 20 December 2021) for the prediction of *cis*-elements and visualization mapping [49]. To identify the binding sites of transcription factors in the promoter region of each *GmHD-Zip III* gene, we searched via the database PlantRegMap (http://plantregmap.cbi.pku.edu.cn/binding_site_prediction.php (accessed on 16 June 2022) with the following parameter: e-value ≤ 1 × 10^−6^ [69]. The visualization of the motifs was performed on the LOGO website (https://weblogo.berkeley.edu/logo.cgi (accessed on 12 June 2022).

### 4.5. Plant Materials and Growth Conditions

The Williams 82 soybean variety was used in this study. The soybean cultivar was grown in a growth room (12 h light, 25 °C, 12 h dark, 22 °C) using a light-emitting diode. The light intensity in the growth chamber was 500 µmol m^−2^ s^−1^.

### 4.6. Tissue Specificity Expression Analysis

Roots, leaves, stems, shoot apical meristems, flowers, pod walls, seeds, and petioles of soybean were collected for tissue-specific expression analysis. The roots, leaves, stems, shoot apical meristems, and petioles were collected from soybean seedlings in the V2 stage (the unfolding of second trifoliate leaves). Flowers were picked three days after flowering. The pod walls and seeds were collected 14 and 30 days after flowering, and each sample was collected thrice independently. Total RNA was extracted from tissues, as described for qRT-PCR. The normalized gene expression values were log2-transformed and visualized in the form of heatmaps constructed using TBtools.

### 4.7. Collection of Leaf Veins Cells

To obtain the high quality RNA of the vascular tissue of soybean plants, the tissues were fixed in Carnoy–acetone fixative (/*v*/*v* = 70/30) for 12 h at 4 °C. Finally, the tissues were transparentized and mounted with t-butyl alcohol. The cells of the cambium, developing xylem, and developing phloem were collected from leaf cross-sections through laser microdissection by using a Veritas Automated Laser Capture Microdissection System (Thermo Fisher Scientific, Inc., Waltham, MA, USA). A minimum of 500 cells were collection from each sample [70,71]. The RNA extracted from laser-captured samples was extracted using the PicoPure RNA Isolation Arcturus Kit. The RNA was amplified to generate cDNA using the Target Amp 2-round aRNA amplification kit for expression analysis [72].

### 4.8. Leaf Anatomy

*GmHD-Zip III* expression was examined during leaf vein development in soybean, and leaf veins were sampled at 0, 12, 24, 48, and 96 h. Vein tissue development was observed by using cross-sections stained with safranin-O/fast green. The 0.5 cm × 1 cm sections were cut from the central leaflet of six different leaves for each treatment for fixation with FAA. Tissues were embedded in paraffin, and serially, paraffin sections were cut into 10 μm thickness. The sections were stained with safranin-O/fast green. Sections were scanned under a Nikon Eclipse 80i microscope, and images were acquired with the ACT-2U imaging software (Nikon Corporation) [73].

### 4.9. Exogenous Hormone Treatments

The following processes were set to elucidate the probable functions of *GmHD-Zip III* in leaf vein development. Seedlings that had been grown for 10 days were sprayed with 100 µmol L^−1^ indole-3-acetic acid (IAA) [45], 100 µmol L^−1^ N-1-naphthylphthalamic acid (NPA) [36], 10 mmol L^−1^ BR, or 5 µmol L^−1^ BRZ [25]. Leaf vein tissues were obtained at 4 h after these treatments. Three individual plant materials were collected, frozen in liquid nitrogen, and stored in a refrigerator at −80 °C. After 10 days of hormone treatment, soybean leaf vein tissue development was observed by using cross-sections stained with safranin-O/fast green. Image J (Image J Software, National Institutes of Health, Bethesda, MD, USA) was employed to measure the areas of xylem, phloem, and cambium of leaf veins. Leaf vein lignin content was quantitatively measured by using the lignin Kit (BC4200, Solarbio, Beijing, China) according to the manufacturer’s protocols.

### 4.10. Expression Analysis

Total RNA was extracted with an RNA prep pure Plant Kit (R6827-01, Omega Bio-tek, Norcross, GA, USA), and the RNA quality was determined by agarose gel electrophoresis [66]. The RNA concentration was determined by NanoDrop^TM^One /One^C^ (Thermo Fisher Scientific Inc., Waltham, MA, USA). gDNA Eraser (TaKaRa Biotechnology, Dalian, China) was used to remove the possible DNA content of the total RNA to avoid detection error when generating the level of gene expression. The first-strand cDNA (Complementary DNA) of RT was synthesized using the PrimeScript^TM^ RT reagent Kit with gDNA Eraser Kit from Takara.

qRT-PCR was conducted on a qRT-PCR system operated on the Quant Studiotm 7 Flex series (Thermo Fisher Scientific Inc., Waltham, MA, USA), using a qRT-PCR Kit (Takara, Dalian, China). The total reaction system had a volume of 10 µL, including 5 µL TBGreen Fastq PCR Mix (2×), 10 µmol L^−1^ upstream and downstream primers (0.4 µL each), and cDNA template (20 ng·μL^−1^), and the total reaction system was 10 µmol L^−1^. The reaction conditions were 95 °C for 3 min, with denaturation at 95 °C for 30 s, annealing at 60 °C for 30 s, and extension at 72 °C for 30 s. A total of 40 PCR cycles were performed. The reaction results obtained by 2^−ΔΔCT^ method were input to QuantStudio^TM^ Real-Time PCR software for analysis [74]. The *GmActin* gene (GLYMA_19G147900) was used as the internal reference gene (Du et al., 2011), and the specific primers are shown in Appendix A. The relative expression values were normalized based on the expression levels of the control. The processed data were log2 transformed, and TBtools software was used to visualize as heatmaps [75].

### 4.11. Statistical Analysis

Statistical analysis was carried out by using one-way ANOVA in SPSS software Version 24 (IBM SPSS Statistics, Armonk, NY, USA). The graphs were drawn by GraphPad Prism 8.0 software (GraphPad Software, Inc., La Jolla, CA, USA).

## 5. Conclusions

We systematically characterized the *GmHD-Zip III* gene family in the soybean genome and the expression profiles of all *GmHD-Zip III* members during leaf vein development in soybean. Twelve *GmHD-Zip III* genes were identified and were found to be unevenly distributed on nine chromosomes. All *GmHD-Zip III* gene family members had similar gene structures and motif arrangements. *GmHB15-L-1* showed stable high expression during leaf vein development and was highly expressed in cambium cells. *GmREV-L-1* and *GmHB14-L-2* maintained stable high expression levels at all xylem developmental stages and were expressed at high levels in vascular cambium cells. Therefore, *GmHB15-L-1* is an essential regulator that is responsible for soybean vein cambial cell formation. *GmREV-L-1* and *GmHB14-L-2* are key components in xylem differentiation. This study will pave the way for research on the different molecular mechanisms of leaf vein development. We will use transient overexpression experiment methods to study the roles of the *HD-Zip III* gene in soybean leaf vein development in the future.

## Figures and Tables

**Figure 1 plants-11-01728-f001:**
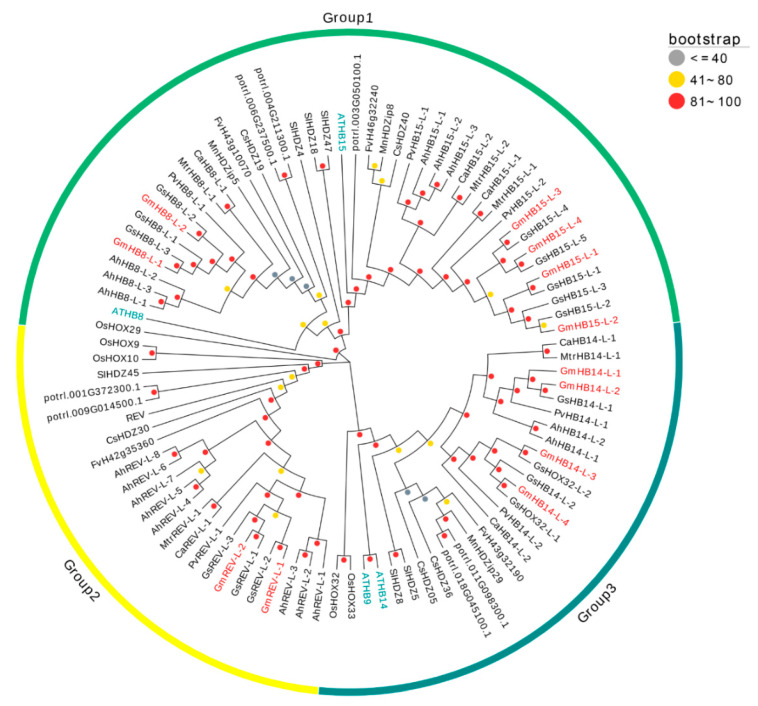
Phylogenetic tree of full-length HD-Zip III proteins from soybean, chickpea, peanut, kidney bean, medicago, wild soybean, strawberry, cucumber, poplar, tomato, rice, and Arabidopsis. The 90 *HD-Zip III* proteins from 13 plant species can be divided into three groups. The words marked in blue are Arabidopsis *HD-Zip III* proteins, and those marked in red are *GmHD-Zip III* proteins. The colors of the circles indicate physical distances: gray represents long distances, and red means short distances.

**Figure 2 plants-11-01728-f002:**
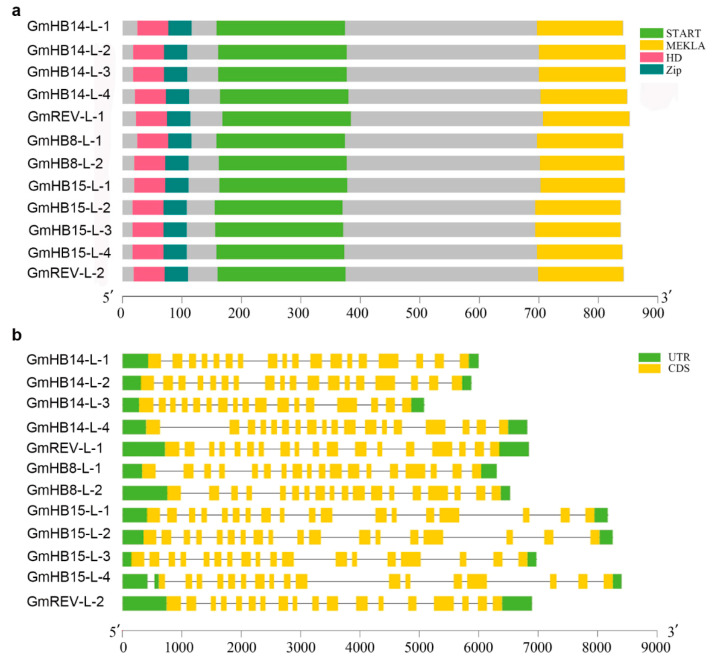
Gene structure, conserved domain, and motif identification. (**a**) Phylogenetic relationships and protein domain prediction of *HD-Zip III* from soybean. The lengths of the proteins and domains can be estimated by using the scale at the bottom. (**b**) The gene structures of the 12 *GmHD-Zip III* genes. Exons and introns are shown as green and yellow boxes, respectively. Gene structures are shown in the right panel.

**Figure 3 plants-11-01728-f003:**
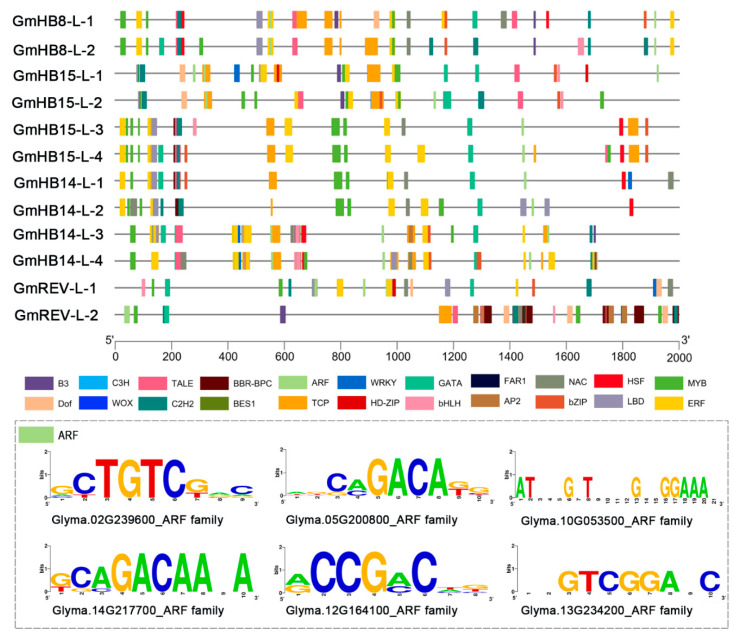
Analysis of *cis*-acting elements in the promoter regions of *GmHD-Zip III* genes.

**Figure 4 plants-11-01728-f004:**
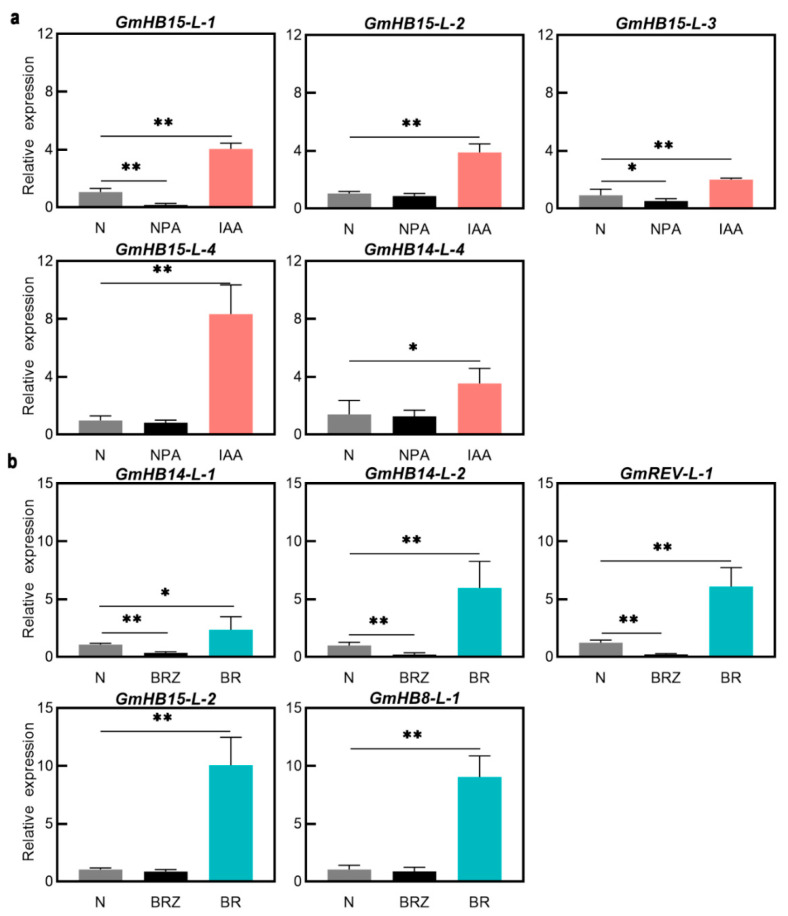
(**a**) Expression profiles of *GmHD-Zip III* genes under IAA and NPA. (**b**) Expression profiles of *GmHD-Zip III* genes under BR and BRZ.Asterisks indicate significant differences between controls and treatments (* *p* < 0.05, ** *p* < 0.01).

**Figure 5 plants-11-01728-f005:**
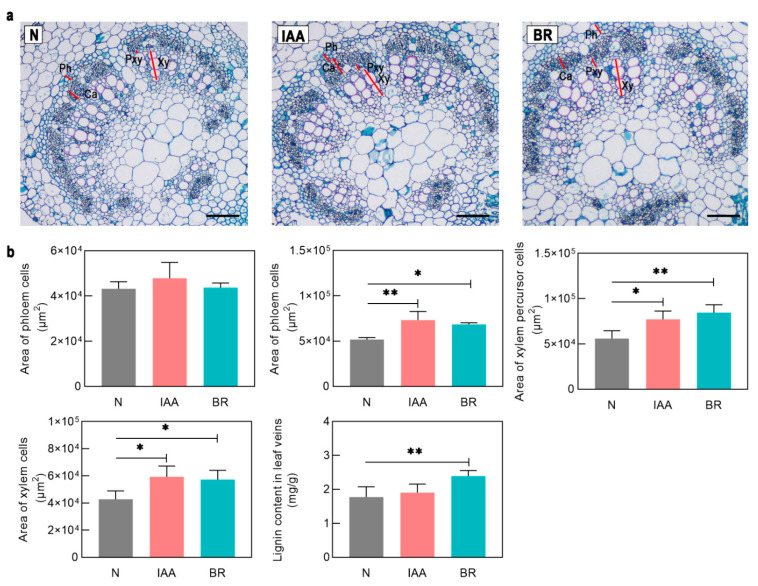
(**a**) The anatomy of the leaf veins. Ca, cambium cell; Ph, phloem cell; TE, tracheary cell; Pxy, xylem precursor cell. All black scale bars indicate 100 μm. (**b**) Phenotypic analysis of soybean leaf veins under IAA and BR treatment. Asterisks indicate significant differences between controls and treatments (* *p* < 0.05, ** *p* < 0.01).

**Figure 6 plants-11-01728-f006:**
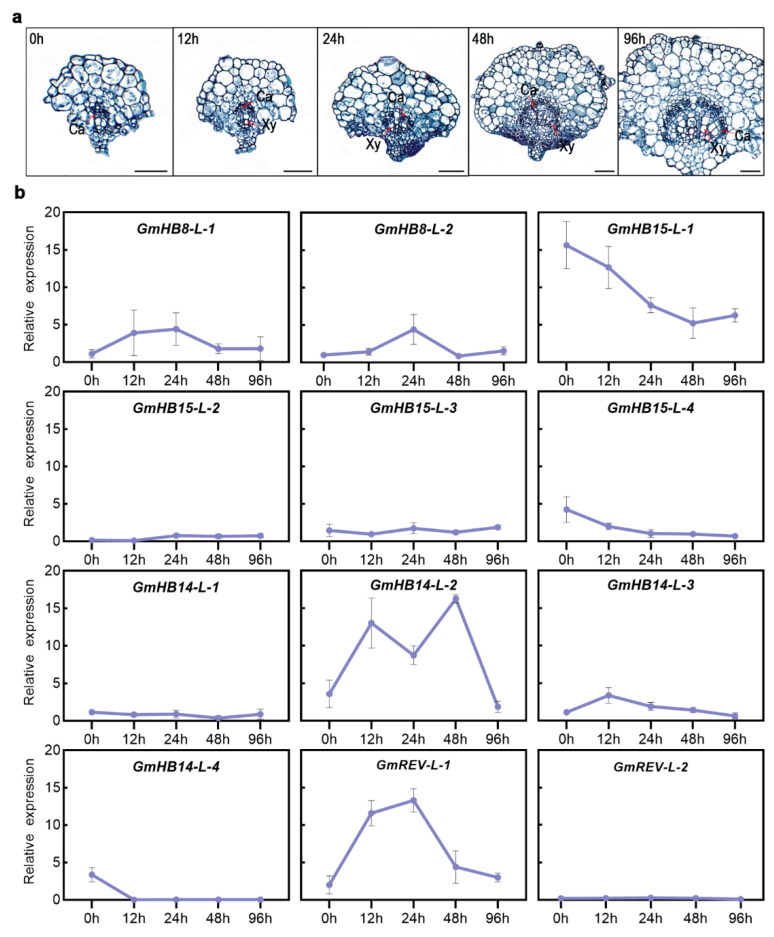
(**a**) Transverse sections of soybean leaf veins in five different developmental stages. All black scale bars indicate 100 μm. (**b**) Expression profiles of 12 *GmHD-Zip III* genes in five stages of vein development. The normalized values of gene expression were log2-transformed and visualized in the form of heatmaps.

**Figure 7 plants-11-01728-f007:**
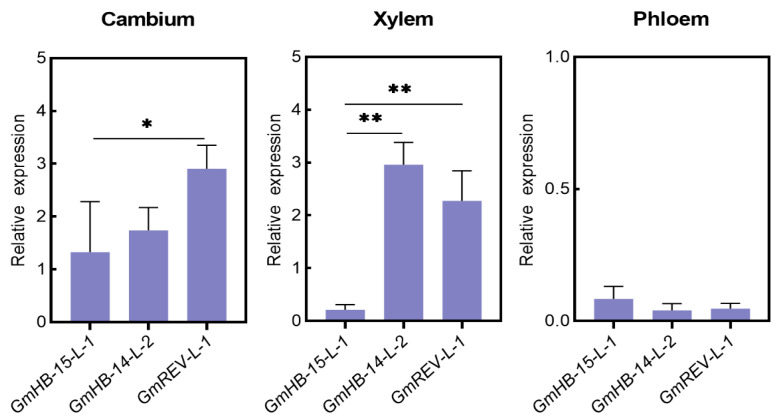
Expression of soybean HD-Zip III genes in various types of vascular cells. Asterisks indicate significant differences between controls and treatments (* *p* < 0.05, ** *p* < 0.01).

## Data Availability

The data presented in this study are available in the article and Appendix A.

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
