# Peer review of "HD-Zip III Gene Family: Identification and Expression Profiles during Leaf Vein Development in Soybean"

_plants, 2022, doi:10.3390/plants11131728_

Round 1
Reviewer 1 Report
"Twelve GmHD-Zip III genes were identified and named" - standalone sentence and no connectivity. The author can improve the flow of information.
"Domain prediction revealed that all of the soybean HD-Zip Ⅲ proteins had four highly conservative domains." - Authors can make this short and simple.
In the introduction, the authors need to further elaborate on the KANADI feedback mechanism and its relevance in vein development.
"A considerable body of evidence" - this expression looks inappropriate here.
In results
"The HD-Zip Ⅲ proteins in soybean can be divided into three 108
groups (Figure 1): GmHB8-L-1, GmHB8-L-2, GmHB15-L-1,
mHB15-L-2, GmHB15-L-3, 109 and GmHB15-L-4 in the first group; GmREV-L-1 and GmREV-L-2 in the second group; 110
and GmHB14-L-1, GmHB14-L-2, GmHB14-L-3, and GmHB14-L-4 in the third group"
This is too descriptive.
Please remove - "The soybean genome contains 20 chromosomes. Their locations in the soybean genome indicated that the 12 GmHD-Zip III genes were not evenly distributed on nine chromosomes and were typically found in gene-dense euchromatic regions near chromosome 114
ends (Figure S1). Chromosomes 4, 5, 6, 9, 11, 12, and 15 contained only one GmHD-Zip Ⅲ gene, whereas chromosome 7 contained two GmHD-Zip Ⅲ genes. Chromosome 8 exhibited the highest density of GmHD-Zip Ⅲ genes." 12 genes can not be present on 20 chromosomes which does not mean the distribution is uneven. This description is pointless.
"cis-Elements in GmHD-Zip Ⅲ promoters" - this does not hold any ground. please condense down this section to one or two sentences only and move the figure to supplementary.
"Figure 3. Expression profiles of GmHD-Zip Ⅲ" - this is very surprising to see a similar pattern of expression for all the analysed genes.
Fig 4- the cross sections are not very clear. Please provide better anatomical images.
Author Response
- Q: "Twelve GmHD-Zip III genes were identified and named" - standalone sentence and no connectivity. The author can improve the flow of information.
A: According to reviewer’s suggestion, we rewrote this section in the course of the revision. In line 19-20, " In this study, twelve HD-Zip III genes were identified from the soybean genome and named. "
- Q: "Domain prediction revealed that all of the soybean HD-Zip Ⅲ proteins had four highly conservative domains." - Authors can make this short and simple.
A: Thanks to the reviewer for this correction. In response to this question, I have completed a rewrite of the Abstract parts of the Manuscript. In line 21-22, "All soybean HD-Zip III proteins contained four highly conserved domains."
- Q: In the introduction, the authors need to further elaborate on the KANADI feedback mechanism and its relevance in vein development.
A: Thanks to the reviewer for the great suggestion. We added new sentences in the Introduction to illustrate this question. In line 64-69, "Loss-of-function KANADI mutants exhibit vascular patterning defects opposite to the rev mutant, increasing xylem cells. HD-Zip III and KANADI have opposing roles in ad/abaxial of the organ formation. Once vascular are formed, HD-Zip III and KANADI are required to coordinate the growth of the adaxial and abaxial."
- Q: "A considerable body of evidence" - this expression looks inappropriate here.
A: We followed this advice and rewrote this part. In line 79, "Auxin is the primary signal involved in the ontogeny of the vascular system."
- Q: In results "The HD-Zip Ⅲ proteins in soybean can be divided into three 108 groups (Figure 1): GmHB8-L-1, GmHB8-L-2, GmHB15-L-1, GmHB15-L-2, GmHB15-L-3, 109 and GmHB15-L-4 in the first group; GmREV-L-1 and GmREV-L-2 in the second group; 110 and GmHB14-L-1, GmHB14-L-2, GmHB14-L-3, and GmHB14-L-4 in the third group" This is too descriptive.
A: Thank you very much. We followed this advice and rewrote this part. In line 124-127, "According to sequence orthologous, 12 GmHD-Zip III genes were divided into three groups. The largest group is Group 1, which contained six GmHD-Zip III genes, while Group 2 and 3 had two and four genes, respectively (Figure 1). "
- Q: Please remove - "The soybean genome contains 20 chromosomes. Their locations in the soybean genome indicated that the 12 GmHD-Zip III genes were not evenly distributed on nine chromosomes and were typically found in gene-dense euchromatic regions near chromosome 114 ends (Figure S1). Chromosomes 4, 5, 6, 9, 11, 12, and 15 contained only one GmHD-Zip Ⅲ gene, whereas chromosome 7 contained two GmHD-Zip Ⅲ genes. Chromosome 8 exhibited the highest density of GmHD-Zip Ⅲ genes." 12 genes can not be present on 20 chromosomes which does not mean the distribution is uneven. This description is pointless.
A: Thank you to the reviewer for pointing this out. We followed this advice and deleted these sentences.
- Q: "cis-Elements in GmHD-Zip Ⅲ promoters" - this does not hold any ground. please condense down this section to one or two sentences only and move the figure to supplementary.
A: Thank you very much. To address this comment, we move the figure in supplemental files. This shortcoming was raised by Reviewer 3 as well. We have used the website that suggested by Reviewer 3 and redrawn Figure 3. In line 186-187, line 196-199, we have rewritten the sentences in the Results sections.
- Q: "Figure 3. Expression profiles of GmHD-Zip Ⅲ" - this is very surprising to see a similar pattern of expression for all the analysed genes.
A: Thanks for your question. Figure 4 shows only the expression patterns of 5 genes significantly induced by IAA and BR. Figure S4 shows the expression profiles of all GmHD-Zip Ⅲ genes after exogenous hormone treatment. To make this figure more reader-friendly, we unified Y-axis in the histograms (Figure 4).
- Q: Fig 4- the cross sections are not very clear. Please provide better anatomical images.
A: Thank you very much. We have redrawn this figure to show the best quality images (Figure 5).
Reviewer 2 Report
Gao et al, have studied the HDZIP TFs in soybean, however, I feel the manuscript is immature at various stages to be credited to publication.
Figures 1 and 2: Authors are interested to study the HDZIP III family of TFs, they have put over the phylogenetic tree and gene information, in my opinion, I cannot find the novel information in these figures to stand alone. Furthermore, various studies have already demonstrated an extensive understanding of that information, for example, please refer, Chen et al. 2014, Belmakar et al. 2014
in Figures 4-5: I cannot understand how the authors have ended up in the soybean leaf vein analysis, under auxin and BR treatments. From the text, this in turn seems to be sudden and I cannot find any firm conclusion from the study.
For me the experiment that is missing and needed is to use VIGS or transient overexpression systems to silence or overexpress the HDZIPs and perform the leaf development analysis possibly through the microscopy analysis, at least this will pertain new information to the readers.
Currently, the manuscript has findings, but it lacks a clear flow of results and novelty to credit a publication.
Author Response
- Q: Figures 1 and 2: Authors are interested to study the HDZIP III family of TFs, they have put over the phylogenetic tree and gene information, in my opinion, I cannot find the novel information in these figures to stand alone. Furthermore, various studies have already demonstrated an extensive understanding of that information, for example, please refer, Chen et al. 2014, Belmakar et al. 2014
A: Thank you to the reviewer for making this point. Genome-wide investigations on the HD-Zip gene family in soybean have been reported, and gene expression in soybean roots under dehydration and salt stress has been studied [1, 2]. Updating the HD-Zip III gene family in soybean has become possible with the release of a new version of the soybean genome. Previous work identified 11 GmHD-Zip Ⅲ genes in the soybean genome (v1.01, JGI Glyma1.0). Compared with Belamkar et al., our study identified one more HD-Zip Ⅲ member, namely, GLYMA_12G075800. All of the soybean HD-Zip Ⅲ proteins we identified belonged to the HD-Zip III family because they contained the four conserved domains of START, MEKHLA, HD, and Zip structure in the HD-Zip III transcription family.
- Q: In Figures 4-5: I cannot understand how the authors have ended up in the soybean leaf vein analysis, under auxin and BR treatments. From the text, this in turn seems to be sudden and I cannot find any firm conclusion from the study.
A: Thank you to the reviewer for making this point. In line 79-99, we added new sentences in the Introduction to illustrate this question. Auxin is involved in the regulation of vascular tissue development [3]. Treatment with a polar auxin transport inhibitor can mimic the vascular defect of HD-Zip Ⅲ mutations [4]. We had treatment with IAA, and the auxin polar transport inhibitor NPA was applied to test whether these genes may play an essential role in auxin-regulated vascular development. HD-Zip III functions in xylem cell differentiation by responding to BR signaling [5]. We analyzed the effects of BRs and BRZ on the accumulation of GmHD-Zip Ⅲ transcripts to investigate the relationship between the expression of these genes and BRs.
- Q: For me the experiment that is missing and needed is to use VIGS or transient overexpression systems to silence or overexpress the HDZIPs and perform the leaf development analysis possibly through the microscopy analysis, at least this will pertain new information to the readers.
A: According to the reviewer’s suggestion, we add related content that the role of the HD-Zip III gene in soybean leaf vein development will be studied using the methods of transient overexpression in the future in the Conclusion.
References
- Belamkar, V., et al., Comprehensive characterization and RNA-Seq profiling of the HD-Zip transcription factor family in soybean (Glycine max) during dehydration and salt stress. BMC Genomics, 2014. 15: p. 950.
- Chen, X., et al., Genome-wide analysis of soybean HD-Zip gene family and expression profiling under salinity and drought treatments. PLoS One, 2014. 9(2): p. e87156.
- Scarpella, E., M. Barkoulas, and M. Tsiantis, Control of Leaf and Vein Development by Auxin. CSH Perspect Biol, 2010. 2(1): p. a001511.
- Zhong, R. and Z.H. Ye, Alteration of auxin polar transport in the Arabidopsis ifl1 mutants. Plant Physiol, 2001. 126(2): p. 549-63.
- Ohashi-Ito, K., T. Demura, and H. Fukuda, Promotion of transcript accumulation of novel Zinnia immature xylem-specific HD-Zip III homeobox genes by brassinosteroids. Plant Cell Physiol, 2002. 43(10): p. 1146-53.
Reviewer 3 Report
The manuscript entitled "HD-Zip III Gene Family: Identification and Expression Profiles
During Leaf Vein Development in Soybean” by Gao and collaborators present a study of a specific type of transcription factors involved in vascular development in Soybean. The study of groups of transcription factors is of great interest, since they are key regulators in different processes. The expression studies presented show the participation of the HD-Zip III group of transcription factors in develomental process. However, it is necessary to clarify different points in the manuscript for publication.
From the point of view of this reviewer, the authors should clarify the following points:
Regarding the identification of the GmHD-Zip III genes, it would be advisable to check if the genes obtained have orthology with the genes obtained by blast. To do this, you can use the PLAZA(https://bioinformatics.psb.ugent.be/plaza/versions/plaza_v4_5_dicots/) database. It would be interesting to study not only the relationship based on homology, but also on orthology.
Regarding the identification of the cis-elements in the promoters, it would be necessary to verify the presence of regulators using other in silico tools, such as PlantRegMap (http://plantregmap.gao-lab.org/). In addition to the most common domains identified, it would also be interesting to present a representation in LOGO (https://weblogo.berkeley.edu/logo.cgi), with the most common bases.
As for Supplementary Figure 4, it is not described in the materials how it has been generated, and from where these data were retrived. Since they have carried out an expression study in different tissues, it would also be interesting to study the expression of the different cis regulators identified in the promoters. This will allow checking if they have a regulatory role on these elements.
From the point of view of this reviewer, the inclusion of these in silico analyzes would improve the study presented in the article.
Author Response
- Q: Regarding the identification of the GmHD-Zip III genes, it would be advisable to check if the genes obtained have orthology with the genes obtained by blast. To do this, you can use the PLAZA(https://bioinformatics.psb.ugent.be/plaza/versions/plaza_v4_5_dicots/) database. It would be interesting to study not only the relationship based on homology, but also on orthology.
A: Thanks to the reviewer for professional suggestions. We have used the PLAZA database to obtain orthology genes (Figure S1). In line 450-451, line 127-130, we added new sentences in the “Methods” and “Result” section.
- Q: Regarding the identification of the cis-elements in the promoters, it would be necessary to verify the presence of regulators using other in silico tools, such as PlantRegMap (http://plantregmap.gao-lab.org/). In addition to the most common domains identified, it would also be interesting to present a representation in LOGO (https://weblogo.berkeley.edu/logo.cgi), with the most common bases.
A: Thanks again to the reviewer for professional suggestions. We followed this advice and redrawn this figure (Figure 3). In line 497-501, line 196-199, we added new sentences in the “Methods” and “Result” section.
- Q: As for Supplementary Figure 4, it is not described in the materials how it has been generated, and from where these data were retrived.
A: Thanks to the helpful comments of the reviewer. Tissue-specific expression of GmHD-Zip Ⅲ was verified by qPCR. The gene expression is visualized in the form of heatmaps constructed using TBtools. In line 508-515, we added detailed experiments in the “Methods” section.
- Q: Since they have carried out an expression study in different tissues, it would also be interesting to study the expression of the different cis regulators identified in the promoters. This will allow checking if they have a regulatory role on these elements.
A: Thank the Reviewer for another fine point. In line 239-241, line 377-384, we added new sentences in the "Result" and "Discussion" sections to illustrate the relationship between cis-elements, expression patterns, and auxin.
Round 2
Reviewer 1 Report
The revised version of the MS looks appropriate.
Reviewer 2 Report
Thank you authors for the revision, please review the language before final accept.
Reviewer 3 Report
Dejar authors,
From the point of view of this reviewer, the introduction of all the suggested changes has improved the manuscript and I consider that it can be published.